# Spousal Concordance in Dietary Behaviors and Metabolic Components, and Their Association: A Cross-Sectional Study

**DOI:** 10.3390/nu12113332

**Published:** 2020-10-29

**Authors:** Dann-Pyng Shih, Chu-Ting Wen, Hsien-Wen Kuo, Wen-Miin Liang, Li-Fan Liu, Chien-Tien Su, Jong-Yi Wang

**Affiliations:** 1Department of Public Health, China Medical University, Taichung City 406040, Taiwan; 90930@cch.org.tw; 2Center for Teaching Excellence, Changhua Christian Hospital, Changhua 50006, Taiwan; 3Strategy Planning Office, Cheng Ching Hospital, Taichung 40764, Taiwan; wenting0310@gmail.com; 4Institute of Environmental and Occupational Health Sciences, National Yang Ming University, Taipei 11221, Taiwan; hwkuo@ym.edu.tw; 5National Defense Medical Center, School of Public Health, Taipei 11490, Taiwan; 6Department of Health Services Administration, China Medical University, Taichung City 406040, Taiwan; wmliang@mail.cmu.edu.tw; 7Institute of Gerontology, College of Medicine, National Cheng Kung University, Tainan 701401, Taiwan; lilian@mail.ncku.edu.tw; 8Department of Family Medicine, Taipei Medical University Hospital, Taipei 110301, Taiwan; ctsu@tmu.edu.tw; 9School of Public Health, College of Public Health, Taipei Medical University, Taipei 110301, Taiwan

**Keywords:** spousal concordance, metabolic components, dietary behavior, fiber food, processed food

## Abstract

This study aims to investigate spousal concordance in dietary behaviors, spousal concordance in metabolic components (MCs), and their association. A cross-sectional survey was conducted in Taiwan from November 2014 to May 2015. Matched-pair analysis, McNemar’s test, logistic regression analysis, and stratified analysis were performed. A total of 901 pairs of spouses (1802 participants) were analyzed. Husbands were less likely to report intakes of high-fiber food (OR_MP_ (matched pairs odds ratio) = 0.30, *p* < 0.0001), fish (OR_MP_ = 0.74, *p* = 0.0128), biscuits or cakes (OR_MP_ = 0.60, *p* < 0.0001), and fast food (OR_MP_ = 0.65, *p* = 0.01) compared with their wives. Husbands had significantly higher odds of being overweight (OR_MP_ = 2.34, *p* < 0.0001); and of having hypertension (OR_MP_ = 2.14, *p* < 0.0001), hypercholesterolemia (OR_MP_ = 1.75, *p* = 0.0007), hyperlipidemia (OR_MP_ = 2.96, *p* < 0.0001), and one or more metabolic components (composite MCs) (OR_MP_ = 2.50, *p* < 0.0001) compared with their wives. After adjusting for age and education, the spousal concordance in high-fiber food intake was inversely associated with the spousal concordance in composite MCs (aOR = 0.62, 95% CI = 0.44–0.88, *p* = 0.0074), whereas the spousal concordance in processed food intake was positively associated with the concordance in composite MCs (aOR (adjusted odds ratio) = 1.56, 95% CI (Confidence Interval) = 1.03–2.36, *p* = 0.034). An intervention study for couples with intakes of different fiber foods and/or processed foods is critical for future study, in order to test what kinds of fiber foods/processed foods are associated with the development of the spousal concordance of metabolic components.

## 1. Introduction

Metabolic syndrome (MS) is an emerging, non-communicable disease (NCD) characterized by the combined presence of several factors that are associated with an increased risk of cardiovascular disease (CVD) and type 2 diabetes mellitus (T2DM), including central obesity, hypertension, hyperlipidemia, elevated fasting plasma glucose, and reduced high-density lipoprotein (HDL) cholesterol [1,2,3]. MS not only leads to morbidity and mortality, but also costs trillions of dollars due to healthcare expenses and the loss of potential economic activity [4,5]. Approximately 10%–50% of the world’s population is suffering from MS, and its prevalence has increased over the past several years [6,7]. MS is the result of interactions between genes, hormones, lifestyle, and the environment; however, the rapid increase in MS is mainly due to shifts in lifestyle and dietary structure to Western patterns, which have been occurring since the 1970s [5,8,9].

Couples often influence each other’s dietary habits and lifestyles. Previous studies have reported that spouses are concordant in several dietary behaviors, such as drinking [10,11,12], alcohol consumption [12,13], fruit and vegetable consumption [14], and the intake of salty food [11]. Furthermore, spousal concordance is also found in MS and its related conditions [15], including being overweight/obesity [16,17], dysglycemia [17], diabetes [18,19,20], hypertension [12,16,21,22], hyperlipidemia [22], coronary heart disease [12,23], and stroke [12]. However, the association between dietary behaviors and metabolic components among spouses is still unclear.

The aim of the present study was to evaluate spousal concordance in food intake behaviors and spousal concordance in metabolic components, and their association. The rationale of investigating the relationship between dietary behaviors and metabolic components among couples in familial pattern studies is that spouses are genetically unrelated, and thus biological confounding factors are avoided.

## 2. Materials and Methods

### 2.1. Data Source

We conducted a cross-sectional study with face-to-face interview surveys at family medicine outpatient clinics in four teaching hospitals in northern, central, and southern Taiwan, from November 2014 to May 2015. A structured questionnaire was designed in order to measure the association between the concordant health behaviors and concordant chronic conditions within households. The questionnaire contained items related to the sociodemographic characteristics of the household members, the relationships among them, self-reported diseases, and health behaviors. The validity of the questionnaire was checked by five experts, including two health professionals (Health Services Administration or health information-related experts) and three professors. The main threshold used to verify the reliability and internal consistency was a Cronbach’s alpha of 0.78.

In this survey, a total of 988 questionnaires were collected, and 25 were excluded for inadequate or missing data. Of the remaining 963 surveys, 901 pairs of spouses (1802 participants) with complete answers were included in the analysis. We defined a couple as “two heterosexual individuals cohabiting in the same house”. This study obtained ethical approval by the Institutional Review Board in the China Medical University Hospital (CMUH103-REC1–116).

### 2.2. Metabolic Components

Metabolic syndrome (MS) is a complex disease defined by the presence of several factors that are associated with an increased risk of cardiovascular disease and type 2 diabetes mellitus. According to the NCEP ATP III (The National Cholesterol Education Program Expert Panel on Detection, Evaluation, and Treatment of High Blood Cholesterol in Adults Treatment Panel III) definition, metabolic components (MCs) include abdominal obesity, high blood pressure, high triglyceride levels, low high-density lipoprotein (HDL) cholesterol levels, and high fasting levels of blood sugar. The definition of metabolic syndrome requires the presence of 3 out of 5 of the metabolic components [1,3].

In the present study, the metabolic components (MCs) included being overweight, hypertension, hypercholesterolemia, and hyperlipidemia. Being overweight was defined as having a body mass index (BMI) ≥ 24 kg/m^2^ (overweight), which is based on the Taiwan HPA (Health Promotion Administration) guidelines (2012) for Taiwanese adult men and women. The cutoff points for hypertension (SPB ≥ 130 mmHg or DBP ≥ 85 mmHg), hypercholesterolemia (total cholesterol (TC) ≥ 200 mg/dL), and hyperlipidemia (Triglyceride (TG) ≥ 200 mg/dL) are applied in clinical settings in Taiwan. Spousal concordance in a metabolic component was defined as being when a husband and wife shared the same metabolic component. A composite metabolic components parameter (composite MCs) was defined as the presence of one or more of the following metabolic components in a participant: being overweight, high blood pressure, high cholesterol, and hyperlipidemia.

### 2.3. Dietary Behaviors

In our study, we developed a set of self-designed questionnaire items, including questions about the intakes of high-fiber food (e.g., vegetables, fruits), fish, red meat (e.g., beef), processed food (e.g., sausage, pickled food), biscuits or cakes, high-fat food, and fast food (e.g., hamburger, fries). The design of these questions was based on health concerns and the pattern of Taiwan’s general dietary behavior. We also referred to the questionnaire of the Nutrition and Health Survey in Taiwan of 2013. The participants were asked a set of questions regarding spousal concordance in dietary behavior. For instance, one question was “Have you had fiber food intake in the past one month?” If the answer was ‘always’ or ‘often’, then this dietary behavior was coded as being present. If the answer was ‘sometimes’, ‘seldom’, or ‘never’, then the dietary behavior was coded as not being present. Spousal concordance in dietary behavior was defined as being when a husband and wife shared a certain food intake behavior. For example, spousal concordance in high-fiber food intake indicates that both spouses reported high-fiber food intake behavior.

### 2.4. Statistical Analysis

The numbers and percentages of the participants’ sociodemographic characteristics (age, education status, dietary behaviors, and metabolic components) were calculated for husbands, wives, and spousal concordance. In order to assess the prevalence rate of concordance in dietary behavior among couples, we cross-tabulated the number (and percentage) of concordant and discordant dietary behaviors among couples in four categories: i.e., both the husband and wife reported the behavior (H + W +); neither the husband nor wife reported the behavior (H − W −); the husband reported the behavior, but the wife did not (H + W −); and the husband did not report the behavior, but the wife did (H − W +). Frequency and matched-pair odds ratio analyses were used to estimate the patterns of spousal concordance in dietary behaviors, and spousal concordance in metabolic components. McNemar’s test was used to determine the differences in food intake behaviors, and the differences in metabolic components between husbands and wives. ‘Paired’ data were used in the estimation of the odds ratios and McNemar’s test. Furthermore, in order to calculate the values for spousal concordance in dietary behaviors and metabolic components, we coded the pattern H + W + as 1 and 0 otherwise. Univariate and multivariate logistic regression analyses were then used to calculate the odds ratios (ORs) and 95% confidence intervals (CIs), and to test the statistical significance of the association between spousal concordance in dietary behaviors and spousal concordance in composite MCs. Stratified analyses using multivariate logistic regression, according to the ages and education of the couples, were used to confirm that the associations between spousal concordance in dietary behaviors and spousal concordance in composite MCs were consistent among different strata. In addition, we also tested the interaction effects between the significant parameters for spousal concordance in dietary behaviors and the adjustment covariates, in order to ensure the appropriateness of the model. All of the multivariate analyses of spousal concordance in the study were adjusted for age, education, and other concordant dietary behaviors among couples. The SAS Software of the System for Windows, version 9.4 (SAS Institute Inc., Cary, NC, USA), was used for the data analyses, and a 2-sided *p* value of 0.05 was considered to be statistically significant.

## 3. Results

### 3.1. Characteristics of the Study Sample

Table 1 shows the sociodemographic characteristics of the participants. A total of 901 pairs of spouses (1802 participants) were included and analyzed in this cross-sectional study. The majority of the participants were older than 50 years (husbands 64.7% and wives 53.1%), had education levels of high school or below (husbands 58.4% and wives 65.6%), and ate high-fiber food (80.2% overall and 68.4% for spousal pairs). The husbands had more metabolic components and related diseases compared to their wives.

### 3.2. Patterns of Dietary Behaviors among Spouses

Table 2 shows the patterns of dietary behaviors among spouses, and we compared the difference between husbands and wives. The top three dietary behaviors that were concordant between spouses (H + W +) were high-fiber food intake (68.4%), fish intake (58.6%), and red meat intake (52.4%). According to the paired-data estimation and McNemar’s test, the husbands were less likely to report intakes of high-fiber food (OR_MP_ = 0.30, *p* < 0.0001), fish (OR_MP_ = 0.74, *p* = 0.0128), biscuits or cakes (matched-pair OR_MP_ = 0.60, *p* < 0.0001), and fast food (OR_MP_ = 0.65, *p* = 0.01) compared with their wives.

### 3.3. Patterns of Metabolic Components among Spouses

Table 3 shows the patterns of metabolic components among spouses, and we compared the differences between husbands and wives. Compared with wives, husbands had a significantly higher odds of being overweight (OR_MP_ = 2.34, *p* < 0.0001), hypertension (OR_MP_ = 2.14, *p* < 0.0001), hypercholesterolemia (OR_MP_ = 1.75, *p* = 0.0007), hyperlipidemia (OR_MP_ = 2.96, *p* < 0.0001), and composite MCs (OR_MP_ = 2.50, *p* < 0.0001).

### 3.4. Association between Concordant Dietary Behaviors and Concordant Composite MCs

Table 4 presents the results of the univariate and multivariate logistic regression analyses. In the multivariate analysis, the likelihood of spousal concordance in composite MCs increased with the wife’s age. The older age of the wife was associated with significantly greater odds of concordance in composite MCs. However, the probability of spousal concordance in composite MCs was inversely related to the husband’s education level. Husbands with high education levels had significantly lower odds of having composite MCs compared with those with low education levels. After adjusting for age, education level, and the other concordant dietary behaviors of couples, spousal concordance in fiber food intake was associated with a 0.62-fold decrease in the odds of spousal concordance in composite MCs (aOR = 0.62, 95% CI = 0.44–0.88, *p* = 0.0074), whereas spousal concordance in processed food intake was associated with a 1.56-fold increase in the odds of spousal concordance in composite MCs (aOR = 1.56, 95% CI = 1.03–2.36, *p* = 0.0340). There were no significant interaction effects between the significant parameters for spousal concordance in dietary behaviors and the adjustment covariates, which supports the appropriateness of the model (results not shown).

### 3.5. Stratified Analysis for the Association between Concordant Dietary Behaviors and Concordant Composite MCs

Figure 1 shows the frequencies of participants with and without concordant high-fiber food intake behaviors, as well as the corresponding events and prevalence rates of concordant composite MCs, according to the ages and education levels of the couples. In all of the analyses, the aORs were less than 1, indicating that the concordant high-fiber food group was negatively associated with concordant composite MCs compared with the non-concordant high-fiber food group. The multivariate logistic regression showed that spousal concordance in high-fiber food intake had significantly negative associations with spousal concordance in composite MCs for husbands aged >50 years (aOR = 0.63, 95% CI = 0.42–0.95, *p* = 0.0271), husbands with an education level of high school or below (aOR = 0.65, 95% CI = 0.43–0.98, *p* = 0.0389), and wives with an education level of high school or below (aOR = 0.62, 95% CI = 0.42–0.90, *p* = 0.0121).

It is noteworthy that the adjusted odds ratios (aORs) and 95% confidence intervals (CIs) for the association between concordant high-fiber food intake behavior and concordant composite MCs were estimated using a multivariate logistic regression adjusted for the husband’s age, the wife’s age, the husband’s education level, the wife’s education level, and all of the concordant dietary behaviors except for concordant high-fiber food intake behaviors. We excluded the covariate that was used to stratify the groups.

Figure 2 shows the frequencies of participants with and without concordant processed food intake behaviors, as well as the corresponding events and prevalence rates of concordant composite MCs, according to the ages and education levels of the couples. Among all of the analyses, the aORs were greater than 1, indicating that the concordant processed food group was significantly associated with a higher likelihood of having concordant composite MCs compared with the non-concordant processed food group. The multivariate logistic regression showed that spousal concordance in processed food intake was associated with 2.01-fold greater odds of spousal concordance in composite MCs for wives aged ≤50 years (aOR = 2.01, 95% CI = 1.03–3.91, *p* = 0.0402).

It is noteworthy that the adjusted odds ratios (aORs) and 95% confidence intervals (CIs) for the association between concordant processed food intake behavior and concordant composite MCs were estimated using a multivariate logistic regression adjusted for the husband’s age, wife’s age, husband’s education level, wife’s education level, and for all of the concordant dietary behaviors except for concordant processed food intake behavior. We excluded the covariate that was used to stratify the groups.

## 4. Discussion

To the best of our knowledge, this is the first study to examine the association between spousal concordance in dietary behaviors and spousal concordance in metabolic components. Using a cross-sectional interview survey, we found that spousal concordance in high-fiber food intake was inversely associated with spousal concordance in composite MCs, whereas spousal concordance in processed food intake was positively associated with a concordance in composite MCs. We also found that husbands were less likely to have fiber food intake, fish intake, biscuit or cake intake, and fast food intake compared to their own wives. The husbands had higher prevalence rates of being overweight, hypertension, hypercholesterolemia, and hyperlipidemia compared to their wives.

This study provides evidence that husbands have approximately 2–3 times higher odds of having metabolic components compared with their wives, including being overweight, hypertension, hypercholesterolemia, and hyperlipidemia. This finding is consistent with the results of previous studies. Several articles have reported a higher prevalence rate in men than in women, especially for those with abdominal obesity and hypertension [24,25,26]. One study is inconsistent with our findings, and reported that the prevalence was higher in women than in men [27]. However, its small sample size might have led to selection bias and unreliable results. In addition, we also found that husbands were less likely to eat high-fiber food and fish compared with their wives. Several studies have demonstrated that a low intake of fiber or fish may result in a higher probability of developing MS, obesity, diabetes, hypertension, hyperlipidemia, CVD, cancer, and other NCDs [6,28,29,30,31,32].

Most of the previous similar studies investigated the spousal concordance pattern, i.e., the husband–wife association, for lifestyle factors and selected diseases separately. Some reported that there was spousal concordance of cancer [33]; depression [34]; mental disorders [35]; physical frailty [36]; and metabolic components, including overweight/obesity [16,17,34], diabetes [18,19,20,34,37], dysglycemia [17,34], and hypertension [12,16,21,22] among couples. However, one Japanese study revealed that spousal concordance in metabolic components was weak and modest [38]. Some studies reported that spouses show concordance in several dietary behaviors, such as drinking [10,11,12], alcohol consumption [12,13], smoking [16,34], fruit or vegetable consumption [14], irregular eating behaviors [34], and eating salty foods [11]. Nevertheless, few addressed the association between concordant lifestyle factors and concordant selected diseases.

Studies on the association between the spousal concordance of dietary behaviors and spousal concordance of metabolic components are uncommon. The main reason may be the complexity of the information. To be specific, the complexity of the study of a husband and wife is different from that of individual research. Due to the differences in the age of the husband and wife, and the time of living together, the progress of the disease has interference in time, and some diseases have genetic conditions. Therefore, if we consider the prevalence rates of the spousal concordance of diseases, the number of some specific diseases might be reduced unexpectedly. In order to solve such problems, we created a composite metabolic components parameter (composite MCs), which was defined according to a less strict standard, compared to the common definition of metabolic syndrome [1,3]. Besides this, we designed a stratified analysis method to verify the stability of the results. Our study indicated that couples that both had the habit of dietary fiber intake were less likely to suffer from the metabolic components concordantly, whereas the spousal concordance of processed food intake was positively associated with the concordance of composite MCs. The results of the stratified analysis were also supportive of these findings.

In our findings, spouses who reported a habit of high-fiber food intake presented around 40% lower odds of being concordant in composite MCs. This result is consistently supported by previous studies. Dietary fiber is found in edible plant foods, such as fruits, vegetables, cereals, and nuts [39]. Dietary fiber can prevent obesity and metabolic syndrome, control blood lipid and blood pressure, and improve glucose levels [30,40]. An increase in dietary fiber intake can reduce the burden of MS risk factors [31]. A possible mechanism for this association could be that high-fiber food can reduce the inflammation generated by the human body. More high-fiber food intake may activate the production of anti-inflammatory cytokines, and may decrease the production of proinflammatory cytokines in the immune system [41]. An animal experiment showed that fermentable fiber (inulin) significantly protected mice against high-fat-diet (HFD)-induced metabolic syndrome by nourishing microbiota to restore interleukin–22 (IL-22), and thereby mediated enterocyte function. The mechanism is that inulin reinstates the HFD-induced loss of enterocyte proliferation, decreases microbiota encroachment, and defends against metabolic syndrome in a microbiota-dependent manner [42].

Our study found that, if both spouses had a habit of processed food intake, then they had about 56% higher odds of being concordant in composite MCs. This finding is consistent with previous studies. Unprocessed food has been found to be a protective factor for MS, whereas processed food, especially ultra-processed food with a high proportion of sugar and saturated fat, is a confirmed risk factor for MS [43,44,45]. A systematic review demonstrated that the high consumption of bread, pizza, hamburgers, processed meat products, snacks and sweets, saturated fat, sodium, and food additives was positively associated with metabolic components [40,45,46]. Chassaing et al. conducted an animal experiment, and concluded that the dietary emulsifiers in processed foods may disturb the balance of gut microbiota and induce gut inflammation, eventually leading to the onset of obesity and metabolic syndrome in mice. The gut microbiota protect the intestine via mucus structures that cover the intestinal surface and separate the gut bacteria from the epithelial cells. The disturbance of the gut microbiota is associated with gut inflammation and metabolic syndrome. The emulsifiers in processed foods can increase the bacterial translocation across epithelia, and result in inflammatory bowel disease. Emulsifier-induced metabolic syndrome has been associated with unbalanced microbiota, species composition alteration, and elevated proinflammatory activity [47].

A future intervention study for couples with intakes of different high-fiber foods and/or processed foods is critical in order to identify the kinds of high-fiber foods/processed foods that are associated with the development of spousal concordance in metabolic components.

There are several limitations in this study. First, the diagnosis of metabolic components was based on the participants’ self-reports. This may result in over-/under-reporting, or misreporting. Self-report biases are widely discussed in behavioral and healthcare research. For example, participants might state that their dietary behavior is healthy, in order to portray themselves in a good light (i.e., social desirability bias). In addition, participants with certain diseases might not report them because they have not been diagnosed. The patterns of over-/under-estimation may vary for different characteristics, such as age, gender, and education level, and they may be affected by the time point in a sequence of interviews [48,49]. We did not adjust for these biases; however, because the validity of the study questionnaire was checked by five experts, and the data were collected at four family medicine outpatient clinics, we believe that the resulting data on dietary behavior and metabolic components are reliable. Second, the absolute data on waist circumference (WC) are not available from this survey. Therefore, we used BMI ≥ 24 kg/m^2^ as one of the metabolic components instead of abdominal obesity, which is suggested by the NCEP ATP III [1,3]. According to the definition from the Ministry of Health and Welfare in Taiwan, BMI ≥ 24 kg/m^2^ is considered to be overweight [50]. BMI and WC are both indirect indicators of body composition, and are commonly used as measurements for adiposity in population studies [51]. Previous studies have demonstrated that BMI and WC perform similarly as indicators of body fatness [52,53], but they are significantly influenced by sex and race [51]. Further study is needed to assess the potential bias of the discrepancy between measurements using BMI ≥ 24 kg/m^2^ and waist circumference. Third, the nature of a cross-sectional study may lead to recall bias, confounding bias, and reporting bias. In order to address these shortcomings, we conducted multivariate logistic regression analyses with adjustments for the ages and education levels of the husbands and wives. Moreover, we performed a stratified analysis by stratifying couples according to their ages and education levels. Overall, our study results were consistent among all of the models. Fourth, some residual confounding factors cannot be ruled out. These factors may be other unmeasured confounding factors or poorly measured confounding factors. For example, physical activity is a potential confounder for metabolic syndrome [15,34], but we did not adjust for this variable due to a lack of sufficient information. Finally, we do not have information about the duration of cohabitation or the timing of the diagnoses of the metabolic components in this study. Therefore, it is not possible to assess the magnitude at which interactions are influenced by the duration of spousal cohabitation. We were also unable to evaluate the severity of metabolic components and their related diseases. Further prospective study is needed in order to take these factors into consideration.

## 5. Conclusions

In conclusion, husbands were less likely to report intakes of high-fiber food, fish, biscuits or cakes, and fast food compared with their wives. Husbands had a higher occurrence of being overweight, hypertension, hypercholesterolemia, and hyperlipidemia compared with their wives. The spousal concordance in high-fiber food intake behavior was inversely associated with spousal concordance in composite metabolic components, whereas spousal concordance in processed food intake behavior was positively associated with spousal concordance in metabolic components. This study suggests that increasing the high-fiber food intake and decreasing the processed food intake can be regarded as two important strategies for the promotion of healthy dietary behaviors among couples in order to reduce the co-occurrence of metabolic components and subsequent diabetes and cardiovascular diseases.

## Figures and Tables

**Figure 1 nutrients-12-03332-f001:**
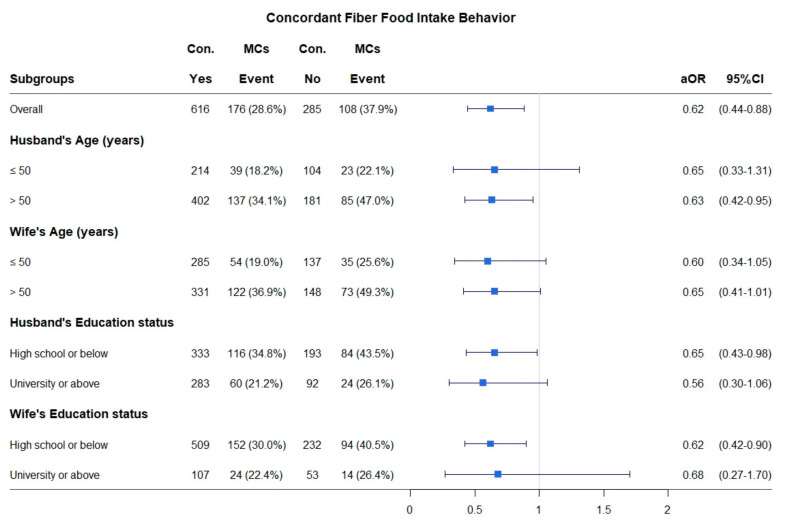
Stratified analysis for the association between concordant high-fiber food intake behavior and concordant composite MCs. Con. MCs: concordant composite metabolic components; ∎: aOR: adjusted odds ratio; CI: confidence interval.

**Figure 2 nutrients-12-03332-f002:**
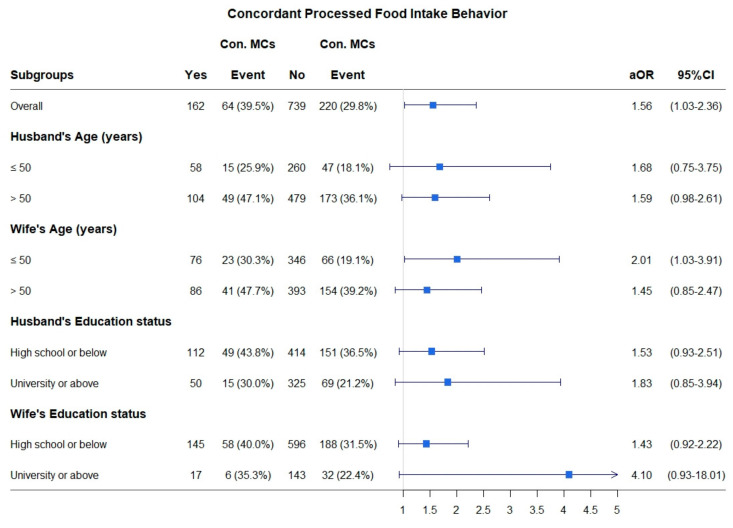
Stratified analysis for the association between concordant processed food intake behavior and concordant composite MCs. Con. MCs: concordant composite metabolic components; ∎: aOR: adjusted odds ratio; CI: confidence interval.

**Table 1 nutrients-12-03332-t001:** Sociodemographic characteristics of the participants (*n* = 901 pairs).

Variables	Husband *n* (%)	Wife *n* (%)	Spousal Concordance * Pairs (%)
**Age (years)**			
≤50	318 (35.3)	422 (46.8)	
51–60	310 (34.4)	284 (31.5)	
>60	273 (30.3)	195 (21.6)	
**Education status**			
High school or below	526 (58.4)	591 (65.6)	
University or above	375 (41.6)	310 (34.4)	
**Dietary behaviors**			
High-fiber food intake	665 (73.8)	781 (86.7)	616 (68.4)
Fish intake	643 (71.4)	684 (75.9)	528 (58.6)
Red meat intake	606 (67.3)	587 (65.2)	472 (52.4)
Processed food intake	285 (31.6)	277 (30.7)	162 (18.0)
Biscuit or cake intake	251 (27.9)	333 (37.0)	127 (14.1)
High-fat food intake	195 (21.6)	216 (24.0)	88 (9.8)
Fast food intake	99 (11.0)	130 (14.4)	42 (4.7)
**Metabolic components**			
Overweight	470 (52.2)	300 (33.3)	173 (19.2)
Hypertension	223 (24.8)	135 (15.0)	58 (2.6)
Hypercholesterolemia	121 (13.4)	79 (8.8)	23 (6.4)
Hyperlipidemia	80 (8.9)	33 (3.7)	9 (1.0)

* Spouses share the same dietary behaviors or metabolic components.

**Table 2 nutrients-12-03332-t002:** Spousal concordance in dietary behaviors, and comparisons among different patterns (*n* = 901 pairs).

Variables	Comparison of Different Patterns among Couples					
	H + W +	H − W −	H + W −	H − W +	OR_MP_	McNemar’s Test
	*n* (%)	*n* (%)	*n* (%)	*n* (%)		*p* Value
High-fiber food intake	616 (68.4)	71 (7.9)	49 (5.4)	165 (18.3)	0.30	<0.0001 ***
Fish intake	528 (58.6)	102 (11.3)	115 (12.8)	156 (17.3)	0.74	0.0128 *
Red meat intake	472 (52.4)	180 (20.0)	134 (14.9)	115 (12.7)	1.17	0.2286
Processed food intake	162 (18.0)	501 (55.6)	123 (13.6)	115 (12.8)	1.07	0.6041
Biscuit or cake intake	127 (14.1)	444 (49.3)	124 (13.8)	206 (22.8)	0.60	<0.0001 ***
High-fat food intake	88 (9.8)	578 (64.1)	107 (11.9)	128 (14.2)	0.84	0.1707
Fast food intake	42 (4.7)	714 (79.2)	57 (6.3)	88 (9.8)	0.65	0.0100 *

OR_MP_: matched pairs odds ratio; * *p* value < 0.05; *** *p* value < 0.001. H + W +: both the husband and wife reported the behavior; H − W −: neither the husband nor the wife reported the behavior; H + W −: the husband reported the behavior, but the wife did not; H − W +: the wife reported the behavior, but the husband did not.

**Table 3 nutrients-12-03332-t003:** Spousal concordance in metabolic components, and comparisons among different patterns (*n* = 901 pairs).

Variables	Comparison of Different Patterns among Couples					
	H + W +	H − W −	H + W −	H − W +	OR_MP_	McNemar’s Test
	*n* (%)	*n* (%)	*n* (%)	*n* (%)		*p* Value
Overweight ^a^	173 (19.2)	304 (33.7)	297 (33.0)	127 (14.1)	2.34	<0.0001 ***
Hypertension	58 (6.4)	601 (66.7)	165 (18.3)	77 (8.6)	2.14	<0.0001 ***
Hypercholesterolemia	23 (2.5)	723 (80.4)	98 (10.9)	56 (6.2)	1.75	0.0007 ***
Hyperlipidemia	9 (1.0)	796 (88.4)	71 (7.9)	24 (2.7)	2.96	<0.0001 ***
Composite Metabolic Components ^b^	284 (31.5)	204 (22.7)	295 (32.7)	118 (13.1)	2.50	<0.0001 ***

OR_MP_: matched pairs odds ratio; *** *p* value < 0.001. H + W +: both the husband and wife reported the behavior; H − W −: neither the husband nor wife reported the behavior; H + W − the husband reported the behavior, but the wife did not; H − W +: the wife reported the behavior, but the husband did not; ^a^: Body Mass Index (BMI) ≥ 24 kg/m^2^; ^b^: composite metabolic components were defined as the presence of one or more of the following metabolic components in a participant: being overweight, hypertension, hypercholesterolemia, and hyperlipidemia.

**Table 4 nutrients-12-03332-t004:** Logistic regression for the associations between concordant dietary behaviors and concordant composite MCs among spouses (*n* = 901 pairs).

Variables	Crude			Multivariate		
	OR	95% CI	*p* Value	aOR	95% CI	*p* Value
**Husband’s Age (years)**						
>50 vs. ≤50 (Reference)	2.54	(1.84, 3.51)	<0.0001 ***	1.49	(0.90, 2.48)	0.1214
**Wife’s Age (years)**						
>50 vs. ≤50 (Reference)	2.57	(1.91, 3.46)	<0.0001 ***	1.80	(1.14, 2.85)	0.0123 *
**Husband’s Education level**						
University or above vs. High school or below (Reference)	0.47	(0.35, 0.64)	<0.0001 ***	0.62	(0.43, 0.88)	0.0074 **
**Wife’s Education level**						
University or above vs. High school or below (Reference)	0.63	(0.42, 0.93)	0.0205 *	1.22	(0.76, 1.97)	0.4138
**Dietary behaviors**						
High-fiber food intake						
Yes vs. No (Reference)	0.66	(0.49, 0.88)	0.0052 **	0.62	(0.44, 0.88)	0.0074 **
Fish intake						
Yes vs. No (Reference)	1.01	(0.76, 1.35)	0.9337	1.17	(0.83, 1.63)	0.3691
Red meat intake						
Yes vs. No (Reference)	0.79	(0.59, 1.04)	0.0911	0.82	(0.59, 1.15)	0.2554
Processed food intake						
Yes vs. No (Reference)	1.54	(1.08, 2.19)	0.0162 *	1.56	(1.03, 2.36)	0.0340 *
Biscuit or cake intake						
Yes vs. No (Reference)	1.00	(0.67, 1.50)	0.9949	1.28	(0.82, 1.99)	0.2762
High-fat food intake						
Yes vs. No (Reference)	1.14	(0.72, 1.81)	0.5850	1.28	(0.73, 2.22)	0.3897
Fast food intake						
Yes vs. No (Reference)	0.76	(0.38, 1.54)	0.4477	0.76	(0.34, 1.69)	0.5021

OR: odds ratio; aOR: adjusted odds ratio; CI: confidence interval. * *p* value < 0.05; ** *p* value < 0.01; *** *p* value < 0.001.

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
