# Peer review of "Spousal Concordance in Dietary Behaviors and Metabolic Components, and Their Association: A Cross-Sectional Study"

_nutrients, 2020, doi:10.3390/nu12113332_

Round 1

Reviewer 1 Report

Some minor grammar typos need addressing. 

Results section difficult to follow.

Limitations mention self-report bias for health indicators, potential for over/under-reporting could be systematically different for reported diet as well as health indicators. 

Author Response

Point 1: Some minor grammar typos need addressing.

Response:

Thank you for the suggestion. We already sent our revised manuscript to MDPI's English editing service. They’ve checked and corrected the grammar and phrasing of our paper.

Point 2: Results section difficult to follow.

Response:

Many thanks for your comment. We revised the Results section and tried to make it clear and easy to read. In addition, we used MDPI's English editing service to improve the grammar and phrasing of our manuscript.

Point 3: Limitations mention self-report bias for health indicators, potential for over/under-reporting could be systematically different for reported diet as well as health indicators.

Response:

Thanks for your advice. We revised our limitation and added new references [48,49] to explain self-report bias and social desirability bias in Lines 310-320.

“There are several limitations in this study. First, the diagnosis of metabolic components was based on the participants’ self-reports. This may result in over-/under-reporting or misreporting. Self-report biases are widely discussed in behavioral and healthcare research. For example, participants might state that their dietary behavior is healthy to portray themselves in a good light (i.e., social desirability bias). In addition, participants with certain diseases might not report them because they have not been diagnosed. The patterns of over-/under-estimation may vary for different characteristics, such as age, gender, and education level, and they may be affected by the time point in a sequence of interviews [48,49]. We did not adjust for these biases; however, because the validity of the study questionnaire was checked by 5 experts and the data were collected at four family medicine outpatient clinics, we believed that the resulting data on dietary behavior and metabolic components are reliable.”

  1. Spitzer, S.; Weber, D. Reporting biases in self-assessed physical and cognitive health status of older Europeans. PloS one 2019, 14, e0223526, doi:10.1371/journal.pone.0223526.
  2. Rosenman, R.; Tennekoon, V.; Hill, L.G. Measuring bias in self-reported data. International journal of behavioural & healthcare research 2011, 2, 320-332, doi:10.1504/ijbhr.2011.043414.

Reviewer 2 Report

Lines 60-63. These are the main results and conclusion of the paper and should not be in the introduction. Please, remove.

Lines 87-89. This is not a standard definition of MS. The authors have not provided justification to use this definition. The ATPIII requires 3 out of 5 components. Here the authors are requiring only one out of the four they have (that are not the standard ones). Not being a standard definition, using the term metabolic syndrome is misleading, and I would suggest to explore the metabolic components that are available separately.  

Also, it is not clear either which cutoffs you are using for high blood pressure, high cholesterol, and hyperlipidemia. Is high cholesterol, total cholesterol? How is hyperlipidemia defined?   Which cutoffs are used for high blood pressure? Finally, the cutoff for BMI should be justified with a reference. I’m okay with using a lower cutoff for Asian populations but this needs to be referenced since it varies depending on the country of origin.

Lines 93-102. Were those the only dietary behaviors that were asked? Or those were the only ones that were analyzed? Was dietary information collected with a validated food frequency questionnaire? Or was it just a bunch of selected ad hoc questions? Please, provide more details.

Line 112. Please, avoid using “effect” in an observational study. Use association instead. Same later on in the paragraph and through the manuscript.

Lines 113-116. If this is analyzed as paired data, why conditional logistic regression (conditioning on the couple) was not performed? I don’t understand well what the outcome is for table 4, is it a binary variable 1 if the couple is concordant for MS and 0 otherwise. Please, describe this in detail because it is not so obvious. I still feel that this should be analyzed as paired data, and therefore with conditional logistic regression. Please, justify otherwise.

Lines 116. “Sensitivity analysis ..”, better saying “Stratified analysis” and maybe later saying that this was a type of sensitivity analysis.

Line 138. Please, do not use the word risk when you are referring to OR. Use odds instead. Here and through the manuscript.

Figures 1 and 2. I wonder why p for interactions were not performed for this stratified analysis. The p value column, if for each OR, is redundant since the CI are already shown. I would suggest to substitute for a p for interaction

The discussion is honestly not very compelling. The purpose of the article is to look at concordance, not if individual dietary behaviors are associated with metabolic components which is kind of known, so part of the discussion can be further summarized.  

Lines 294. There were many important potential confounders that were not adjusted for. For example, physical activity. This is a huge limitation of this study.

The manuscript will benefit from an English editor review.

Author Response

Point 1: Lines 60-63. These are the main results and conclusion of the paper and should not be in the introduction. Please, remove.

Response:

Thanks for your comment. Lines 60-63 have been removed accordingly.

Point 2: Lines 87-89. This is not a standard definition of MS. The authors have not provided justification to use this definition. The ATPIII requires 3 out of 5 components. Here the authors are requiring only one out of the four they have (that are not the standard ones). Not being a standard definition, using the term metabolic syndrome is misleading, and I would suggest to explore the metabolic components that are available separately.

Response:

Thanks for your critical and valuable advice. We changed the term “metabolic syndrome” to “metabolic components” in the title and through the manuscript as per your suggestion. In addition, we revised our methods with more clear definitions about metabolic syndrome (MS), metabolic components (MCs), and composite metabolic components (composite MCs). Please refer to Line 84-99.

“Metabolic syndrome (MS) is a complex disease defined by the presence of several factors that are associated with an increased risk of cardiovascular disease and type 2 diabetes mellitus. According to the NCEP ATP III definition, metabolic components (MCs) include abdominal obesity, high blood pressure, high triglyceride levels, low high-density lipoprotein (HDL) cholesterol levels, and high fasting levels of blood sugar. The definition of metabolic syndrome requires the presence of 3 out of 5 metabolic components [1,3].

In the present study, the metabolic components (MCs) included overweight, hypertension, hypercholesterolemia, and hyperlipidemia. Overweight was defined as body mass index (BMI) ≧ 24 kg/m2 (overweight), which is based on Taiwan HPA guidelines (2012) for Taiwanese adult men and women. The cutoff points for hypertension (SPB≧130mmHg or DBP≧85mmHg), hypercholesterolemia (total cholesterol (TC) ≧ 200 mg/dl), and hyperlipidemia (Triglyceride (TG) ≧200 mg/dl) are applied in clinical setting in Taiwan. Spousal concordance in a metabolic component was defined as a husband and wife who shared the same metabolic component. A composite metabolic components parameter (composite MCs) was defined as the presence of more than one of the following metabolic components in a participant: overweight, high blood pressure, high cholesterol, and hyperlipidemia.”

Point 3: Also, it is not clear either which cutoffs you are using for high blood pressure, high cholesterol, and hyperlipidemia. Is high cholesterol, total cholesterol? How is hyperlipidemia defined?  Which cutoffs are used for high blood pressure? Finally, the cutoff for BMI should be justified with a reference. I’m okay with using a lower cutoff for Asian populations but this needs to be referenced since it varies depending on the country of origin.

Response:

Thanks for your comment. We have revised this section. Please refer to Line 90-95.

“In the present study, the metabolic components (MCs) included overweight, hypertension, hypercholesterolemia, and hyperlipidemia. Overweight was defined as body mass index (BMI) ≧ 24 kg/m2 (overweight), which is based on Taiwan HPA guidelines (2012) for Taiwanese adult men and women. The cutoff points for hypertension (SPB≧130mmHg or DBP≧85mmHg), hypercholesterolemia (total cholesterol (TC) ≧ 200 mg/dl), and hyperlipidemia (Triglyceride (TG) ≧200 mg/dl) are applied in clinical setting in Taiwan.”

Point 4: Lines 93-102. Were those the only dietary behaviors that were asked? Or those were the only ones that were analyzed? Was dietary information collected with a validated food frequency questionnaire? Or was it just a bunch of selected ad hoc questions? Please, provide more details.

Response:

Thanks for your comment. We have revised this section. Please refer to Line 100-112.

“2.3. Dietary Behaviors

In our study, we developed a set of self-designed questionnaire items, including questions about the intakes of high-fiber food (e.g., vegetables, fruits), fish, red meat (e.g., beef), processed food (e.g., sausage, pickled food), biscuits or cakes, high-fat food, and fast food (e.g., hamburger, fries). The design of these questions was based on health concerns and the pattern of Taiwan’s general dietary behavior. We also refer the questionnaire of the Nutrition and Health Survey in Taiwan of the 2013. The participants were asked a set of questions regarding spousal concordance in dietary behavior. For instance, one question is, “Have you had fiber food intake in the past one month?” If the answer was “always” or “often”, then this dietary behavior was coded as being present. If the answer was “sometimes”, “seldom”, or “never”, then the dietary behavior was coded as not being present. Spousal concordance in dietary behavior was defined as a husband and wife who shared certain food intake behavior. For example, spousal concordance in high-fiber food intake indicates that both spouses reported high-fiber food intake behavior.”

Point 5: Line 112. Please, avoid using “effect” in an observational study. Use association instead. Same later on in the paragraph and through the manuscript.

Response:

Thanks for your critical and important advice. We revised our manuscript accordingly. The term “effect” had been removed and we used “association” instead of through the manuscript.

Point 6: Lines 113-116. If this is analyzed as paired data, why conditional logistic regression (conditioning on the couple) was not performed? I don’t understand well what the outcome is for table 4, is it a binary variable 1 if the couple is concordant for MS and 0 otherwise. Please, describe this in detail because it is not so obvious. I still feel that this should be analyzed as paired data, and therefore with conditional logistic regression. Please, justify otherwise.

Response:

Many thanks for your comment. We revised this part and explained in Lines 125-129.

“Furthermore, to calculate values for spousal concordance in dietary behaviors and metabolic components, we coded the pattern H+W+ as 1 and 0 otherwise. Univariate and multivariate logistic regression analyses were then used to calculate the odds ratios (ORs) and 95% confidence intervals (CIs) and test the statistical significance of the association between spousal concordance in dietary behaviors and spousal concordance in composite MCs.”

Point 7: Lines 116. “Sensitivity analysis ..”, better saying “Stratified analysis” and maybe later saying that this was a type of sensitivity analysis.

Response:

Thanks for your comments. We revised our manuscript accordingly. The word sensitivity had used stratified instead.

Point 8: Line 138. Please, do not use the word risk when you are referring to OR. Use odds instead. Here and through the manuscript.

Response:

Thanks for your critical and important advice. We revised our manuscript accordingly. The word “risk” had been removed and used odds instead.

Point 9: Figures 1 and 2. I wonder why p for interactions were not performed for this stratified analysis. The p value column, if for each OR, is redundant since the CI are already shown. I would suggest to substitute for a p for interaction

Response:

Thanks for your critical and important advice. We followed your advice to analyze the interaction effect and found the p-value all greater than 0.05. We added this finding in the results section. Please refer to Line 190-192.

“There were no significant interaction effects between significant parameters for spousal concordance in dietary behaviors and the adjustment covariates, which supports the appropriateness of the model (results not shown).

Our original thinking was tried to stratify the sample into subgroups and investigate whether the results of association were stable among different subgroups. Therefore, we performed the stratified analysis. We agreed that we do not need to present the p-value in the figures. Hence, we removed the p-value from figure 1 and figure 2.

Point 10: The discussion is honestly not very compelling. The purpose of the article is to look at concordance, not if individual dietary behaviors are associated with metabolic components which is kind of known, so part of the discussion can be further summarized.”

Response:

Many thanks for your advice. We revised our manuscript with less individual dietary behavior and more spousal concordance in metabolic components (Lines 254-277) in the discussion.

“Most of the previous similar studies investigated the spousal concordance pattern, i.e., husband-wife association, for lifestyle factors and selected diseases separately. Some reported that there was spousal concordance of cancer [33], depression [34], mental disorder [35], physical frailty [36], metabolic components, including overweight/obesity [16,17,34], diabetes [18-20,34,37], dysglycemia [17,34], and hypertension [12,16,21,22] among couples. However, one Japanese study revealed that spousal concordance in metabolic components was weak and modest [38]. Some studies reported that spouses show concordance in several dietary behaviors, such as drinking [10-12], alcohol consumption [12,13], smoking [16,34], fruit or vegetable consumption [14], irregular eating behavior [34] and eating salty foods [11]. Nevertheless, few addressed the association between concordant lifestyle factors and concordant selected diseases.

Studies on the association between spousal concordance of dietary behaviors and spousal concordance of metabolic components are uncommon. The main reason may be the complexity of the information. To be specific, the complexity of the study of husband and wife is different from that of individual research. Due to the differences in the age of husband and wife and the time of living together, the progress of the disease has interference in time, and some diseases have genetic conditions. Therefore, if we consider the prevalence rates of spousal concordance of diseases, the number of some specific diseases might be reduced unexpectedly. In order to solve such problems, we created a composite metabolic components parameter (Composite MCs), which was defined according to a less strict standard, compared to the common definition of metabolic syndrome [1,3]. Besides, we designed a stratified analysis method to verify the stability of the results. Our study indicated that couples that both had the habit of dietary fiber intake were less likely to suffer from the metabolic components concordantly, whereas spousal concordance of processed food intake was positively associated with the concordance of composite MCs. The results of the stratified analysis were also supportive of these findings.”

Point 11: Lines 294. There were many important potential confounders that were not adjusted for. For example, physical activity. This is a huge limitation of this study.

Response:

Thanks for your comment. We revised our limitation and added physical activity as a potential confounder with two citations. Please refer to Line 333-334.

“Fourth, physical activity is a potential confounder for metabolic syndrome [15,34], but we did not adjust for this variable due to a lack of sufficient information.”

Point 12: The manuscript will benefit from an English editor review.

Response:

Thank you for the suggestion. We already sent our revised manuscript to MDPI's English editing service. They’ve checked and corrected the grammar and phrasing of our paper.

Reviewer 3 Report

Dear Authors,

The manuscript describes the effect of spousal concordance of dietary behaviors on spousal concordance of metabolic syndrome based on data derived from a cross-sectional survey. Statistical analysis involves McNemar’s test, frequency analysis, logistic regression analysis extended by sensitivity analysis.  The authors conclude that spousal concordance of fiber food intake and spousal concordance of processed food intake may affect the risk of MS's spousal concordance. The manuscript provides interesting findings, but

I have only some minor considerations:

1) Review the English language

2) Check the quality of the images presented in the manuscript  (see Figure 1 and Figure 2; should be at the same quality)

3) Missing backspace between 901pairs in titles of some Tables

4) Suggestion: Variables in a column: Education status, dietary behaviors, metabolic syndrome components should be highlighted by italic style or separated by an additional empty line to be more visible in Table 1.

5) Consider replacing Husband and wife both had etc. for shortcuts H+W+, H+W-,  H+W+, etc. to make the pattern of dietary behavior more informative for readers (Table 2 and Table 3).

6) Authors concluded that the intervention for wife’s dietary behavior change would improve husband’s health and prevent MS. Authors also wrote that wife could influence husbands’ dietary behavior because wives are in charge of family food selection and preparation [46,47] (citations are from 1992 and 1999).

Please consider here that today husband can decide about his diet itself and, such as his wife, maybe and even should be actively engaged in preferring healthy dietary behavior.

7) In the discussion part of the manuscript, please briefly (2-3 sentences) explain the mechanism of fiber action on microbiota to restore interleukin-22 (IL-22) mediated enterocyte function.

8) Also, in the discussion section, please briefly (2-3 sentences) explain how processed foods act to disturb the balance of gut microbiota.

Author Response

The manuscript describes the effect of spousal concordance of dietary behaviors on spousal concordance of metabolic syndrome based on data derived from a cross-sectional survey. Statistical analysis involves McNemar’s test, frequency analysis, logistic regression analysis extended by sensitivity analysis.  The authors conclude that spousal concordance of fiber food intake and spousal concordance of processed food intake may affect the risk of MS's spousal concordance. The manuscript provides interesting findings, but

I have only some minor considerations:

Point 1: Review the English language

Response:

Thank you for the suggestion. We already sent our revised manuscript to MDPI's English editing service. They’ve checked and corrected the grammar and phrasing of our paper.

Point 2: Check the quality of the images presented in the manuscript (see Figure 1 and Figure 2; should be at the same quality)

Response:

Thank you for your comment. We already changed the new Figure 1 and Figure 2 in the revised manuscript.

Point 3: Missing backspace between 901pairs in titles of some Tables

Response:

Thank you very much for this comment. We added backspaces between 901 pairs in titles of Tables accordingly.

Point 4: Suggestion: Variables in a column: Education status, dietary behaviors, metabolic syndrome components should be highlighted by italic style or separated by an additional empty line to be more visible in Table 1.

Response:

Thanks for your suggestion. We already revised and highlighted age, education status, dietary behaviors, metabolic components by bold to make it more visible.

Point 5: Consider replacing Husband and wife both had etc. for shortcuts H+W+, H+W-, H+W+, etc. to make the pattern of dietary behavior more informative for readers (Table 2 and Table 3).

Response:

Thanks for your suggestion. We already replaced for Table 2, lines 163-168, Table 3 and lines 169-177 in the revised manuscript.

Point 6: Authors concluded that the intervention for wife’s dietary behavior change would improve husband’s health and prevent MS. Authors also wrote that wife could influence husbands’ dietary behavior because wives are in charge of family food selection and preparation [46,47] (citations are from 1992 and 1999).

Please consider here that today husband can decide about his diet itself and, such as his wife, maybe and even should be actively engaged in preferring healthy dietary behavior.

Response:

Thanks for your comment. This part has been removed according to your suggestion.

Point 7: In the discussion part of the manuscript, please briefly (2-3 sentences) explain the mechanism of fiber action on microbiota to restore interleukin-22 (IL-22) mediated enterocyte function.

Response:

Thanks for your comment. We revised this part with the mechanism of fiber action in Lines 286-291.

“An animal experiment showed that fermentable fiber (inulin) significantly protected mice against high-fat-diet (HFD)-induced metabolic syndrome by nourishing microbiota to restore interleukin-22 (IL-22) and thereby mediated enterocyte function. The mechanism is that inulin reinstates the HFD-induced loss of enterocyte proliferation, decreases microbiota encroachment, and defends against metabolic syndrome in a microbiota-dependent manner [42].”

Point 8: Also, in the discussion section, please briefly (2-3 sentences) explain how processed foods act to disturb the balance of gut microbiota.

Response:

Many thanks for your suggestion. We revised this part and explained how processed foods act to disturb the balance of gut microbiota in Line 298-306.

“Chassaing et al. conducted an animal experiment and concluded that dietary emulsifiers in processed foods may disturb the balance of gut microbiota and induce gut inflammation, eventually leading to the onset of obesity and metabolic syndrome in mice. Gut microbiota protects the intestine via mucus structures that cover the intestinal surface and separate gut bacteria from epithelial cells. The disturbance of gut microbiota is associated with gut inflammation and metabolic syndrome. Emulsifiers in processed foods can increase bacterial translocation across epithelia and result in inflammatory bowel disease. Emulsifier-induced metabolic syndrome has been associated with unbalanced microbiota, species composition alteration, and elevated proinflammatory activity [47].”

Round 2

Reviewer 2 Report

The authors have been very responsive to this reviewer's comments and the manuscript has improved in clarity. I only have two minor comments:

Line 97. Please change to “two or more”. It’s more in the line to the standard definitions (ATPIII says 3 or more, not more than 2)

Lines 334. On top of physical activity there may be other unmeasured confounders or poorly measured confounders, so please, add that residual confounding cannot be ruled out

Author Response

Response to Reviewer 2 Comments

Comments and Suggestions for Authors

The authors have been very responsive to this reviewer's comments and the manuscript has improved in clarity. I only have two minor comments:

Point 1: Line 97. Please change to “two or more”. It’s more in the line to the standard definitions (ATPIII says 3 or more, not more than 2)

Response:
Thanks for your comment. We revised our manuscript accordingly. Please see Lines 30, 97, 176.

Line 96-98:
A composite metabolic components parameter (composite MCs) was defined as the presence of one or more of the following metabolic components in a participant: overweight, high blood pressure, high cholesterol, and hyperlipidemia.

Point 2: Lines 334. On top of physical activity there may be other unmeasured confounders or poorly measured confounders, so please, add that residual confounding cannot be ruled out

Response:

Thanks for your comment. We revised our manuscript accordingly. Please see Lines 333-334.

Fourth, some residual confounding factors cannot be ruled out. These factors may be other unmeasured confounding factors or poorly measured confounding factors. For example, physical activity is a potential confounder for metabolic syndrome [15,25]. However, we did not adjust this variable due to a lack of sufficient information.
